# Technical note: Isolating methane emissions from animal feeding operations in an interfering location

Megan E. McCabe[1], Ilana B. Pollack[2], Emily V. Fischer[2], Kathryn M. Steinmann[1], and Dana R. Caulton[1]

[1]Department of Atmospheric Science, University of Wyoming, Laramie, WY, 82071, USA
[2]Department of Atmospheric Science, Colorado State University, Fort Collins, Colorado, 80523, USA

**Correspondence:** Dana R. Caulton (dcaulton@uwyo.edu)

**Abstract.** Agriculture emissions, including those from cattle and dairy concentrated animal feeding operations (CAFOs), make up a large portion of the United States' total greenhouse gas emissions. However, many CAFOs reside in areas where methane ($CH_4$) from oil and natural gas complicates the quantification of CAFO emissions. Traditional approaches to quantify emissions in such regions often relied on inventory subtraction of other known sources. We compare the results of two approaches to attribute CAFOs $CH_4$ emission rate from an aircraft mass-balance derived $CH_4$ emission rate. These methods make use of the $CH_4$, ethane ($C_2H_6$) and ammonia ($NH_3$) mixing ratio data collected simultaneously in-flight downwind of CAFOs in northeastern Colorado. The first approach, subtraction method, is similar to inventory subtraction except the amount to be removed is derived from the observed $C_2H_6$ to $CH_4$ ratio rather than an inventory estimate. The results from this approach showed high uncertainty, primarily due to how error propagates through subtraction. Alternatively, multivariate regression (MVR) can be used to estimate CAFO $CH_4$ emissions using the $NH_3$ emission rate and an $NH_3$ to $CH_4$ ratio. These results showed significantly less uncertainty. We identified criteria to determine the best attribution method; these criteria can support attribution in other regions. The final emissions estimates for the CAFO presented here were 13 ($\pm$3) g $CH_4$ head$^{-1}$ hr$^{-1}$ and 13 ($\pm$2) g $NH_3$ head$^{-1}$ hr$^{-1}$. These estimates are higher than the US EPA inventory and previous studies highlighting the need for more measurements of $CH_4$ and $NH_3$ emission rates.

## 1 Introduction

Livestock produce large amounts of greenhouse gases (GHG) and reactive nitrogen species, including methane ($CH_4$) and ammonia ($NH_3$), through enteric fermentation and waste generation. Ruminant animals (e.g. cattle, buffalo, sheep, goats, and camels) constitute a significant source of $CH_4$ as their digestive systems break down coarse plant material through microbial fermentation in their rumen stomach (large frontal stomach) and subsequently release the produced gas ($CH_4$). From 1990 to 2019, $CH_4$ emissions from enteric fermentation grew 8.4% in the United States, making agriculture the largest source of US $CH_4$ anthropogenic emission in 2020 (EPA, 2022). Waste and manure management are also significant emission sources of $CH_4$ and $NH_3$. Together enteric fermentation and manure management account for more than 30% of US anthropogenic $CH_4$

emissions (Maasakkers et al., 2016; EPA, 2022). $CH_4$ is a crucial GHG due to its high global warming potential (Myhre et al., 2013; Moumen et al., 2016; Smith et al., 2021).

Large uncertainty remains around the magnitude of livestock emissions. Beef and dairy cattle are confined to feedlots where they are fed and kept in tight areas to process the animals efficiently. These feedlots are known as concentrated animal feeding operations (CAFOs). The compactness of CAFOs has been shown to create significant emissions (Golston et al., 2020; Hacker et al., 2016; Eilerman et al., 2016; Staebler et al., 2009). However, observations indicated large variability in these $CH_4$ emissions, which creates uncertainty in cumulative estimates of agricultural emissions (Golston et al., 2020). There are many

different factors determining the amount of $CH_4$ released from enteric fermentation and manure management and practices may vary from farm to farm (EPA, 2022; Maasakkers et al., 2016). However, interfering sources of $CH_4$ and $NH_3$, such as oil and natural gas (ONG), waste pools, and landfills, etc., may also complicate measurements from individual CAFOs. Improving the methodology for isolating CAFO signals from other interfering sources will allow more accurate measurements and provide new information to constrain greenhouse gas emissions from the agriculture sector.

When $CH_4$ concentration data are used alone for emissions quantification in regions with multiple $CH_4$ sources there is not enough information to distinguish the contributing sources. However, previous studies have attributed $CH_4$ emissions in complicated regions using a variety of methods. A few examples include attribution by subtracting inventory data (Caulton et al., 2014; Peischl et al., 2015, 2018), collecting ground based isotope data to attribute $CH_4$ signals (Townsend-Small et al., 2016), using $C_2H_6$ as a tracer to subtract or attribute the ONG fraction, (Mielke-Maday et al., 2019), and using multivariate

regression (MVR) using independent gas tracers to attribute sectors (Kille et al., 2019; Pollack et al., 2022).

    There are many considerations when determining which attribution method is most appropriate, and often attribution is done based on the data available rather than the ideal methodology. Attribution using the subtraction of a value determined by inventory or calculated ratio is best done for larger regions due to the variability of emissions from individual sources. For ONG, individual sites can release different amounts of $CH_4$ and ethane ($C_2H_6$) compared to other wells and compressors

(Yacovitch et al., 2014; Zimmerle et al., 2022). When quantifying emissions in a small region (as in this work), an estimate of emissions from each type of ONG source or a local $C_2H_6$:$CH_4$ ratio near the CAFO is required to accurately separate the $CH_4$ emissions into contributions from the CAFO and nearby ONG activities.

    Another concern with using a tracer like $C_2H_6$ in isolation to estimate the contribution of ONG is that in complicated regions individual $CH_4$ signals become mixed, making it possible that the observed ratio by aircraft is not representative of

the original ratio at the ground. This is the theory behind tracer release, for example, where a tracer gas is released at a known rate near a source of interest and used to back out the source emission rate (Roscioli et al., 2015). The added gas does not have to be introduced exactly at the source emission point (which may be unknown), provided sampling occurs far enough downwind where the species are well mixed. Typically this means the tracer gas must be released within 100 m of the source and measured >500 m downwind (Roscioli et al., 2015). For airborne data there are many situations where we would

expect signals to be mixed complicating the use of airborne ratio analysis. Townsend-Small et al. (2016) circumvented this by combining ground-based isotope ratios (which show distinct ratios for particular sources) with aircraft data (which showed only one ratio). However, because they used a single isotope ratio there is large uncertainty in their results stemming from the

single isotope ratio that must be attributed to multiple contribution signals (1 equation, multiple unknowns). This is similar to the difficulties that Smith et al. (2015) encountered using $C_2H_6$ as a tracer in a complicated region with multiple $C_2H_6$:$CH_4$ ratios.

On the other hand, an approach like MVR, which makes use of multiple tracer gases, requires sufficient data and is subject to its own sensitivities (Kille et al., 2019). MVR is best used when there are multiple tracer gases that can be treated as independent variables and one dependent variable (i.e. $CH_4$). The more independent variables that are included, the more data is needed to produce statistically significant results. These independent variables can be thought of as independent source terms. Although MVR is generally an effective method, it is important to note that it may not be appropriate in all situations. Specifically, using MVR with too many independent variables in a region where there are only a few sources may result in misleading or inaccurate results. However, using MVR with a limited number of independent variables, corresponding to known sources can yield reliable results for those specific sources. Even in cases where there is a possibility of additional sources, fewer independent variables will produce more robust results and other sources can be treated in the extra term/ background.

Here we demonstrate a methodology to isolate and quantify emission rates for CAFOs in the northeastern Colorado Front Range (NCFR) using airborne measurements of $CH_4$, $C_2H_6$, and $NH_3$. We investigated two methods for CAFO $CH_4$ emission isolation: 1) a subtraction method using the $C_2H_6$:$CH_4$ ratio and 2) a MVR method using $CH_4$, $C_2H_6$ and $NH_3$. This study focuses on the NCFR, where there is a high density of large CAFOs. Figure 1 shows a map of the NCFR with CAFOs for beef cattle, dairy cattle, chickens, sheep, and swine in the area. The area is dominated by beef cattle and dairy CAFOs. However, the NCFR contains a large mixture of $CH_4$ emissions, due to the high production of ONG. In this region, prior estimates indicate that natural gas accounts for 38.5% of the state-wide $CH_4$ emissions, while agriculture accounts for 22.3% of the state-wide $CH_4$ emissions (Arnold et al., 2014). The NCFR is home to Denver-Julesburg Basin (DJB), with over 52,000 ONG wells, and has an abundance of compressors and processing plants; many of which are in close proximity to CAFOs (Higley and Cox, 2007).

## 2 Materials and Methods

### 2.1 Data Collection

This study was conducted in the NCFR near Greeley, CO; the analysis includes data collected over Weld, Morgan, Logan, Larimer, and Washington counties. There are many CAFOs within these five counties with an area wide maximum capacity >1,000,000 heads of cattle (United States Department of Agriculture [USDA], 2018). This study used the University of Wyoming King Air (UWKA), which is a relatively small research aircraft capable of flying at low altitudes (100 m AGL) and slow flight speeds (95 m/s). The UWKA is a national aircraft research facility owned and operated by the University of Wyoming. Flights departed from and returned to the Laramie Airport in Laramie, Wyoming (KLAR). Data outside of the NCFR was removed to ensure regional data only. This was done by screening everything out north and west of 41° latitude, -105.25° longitude. Three flights were performed in November 2019, departing around 12:00 Mountain Standard Time (MST) and lasting 2-4 hours.

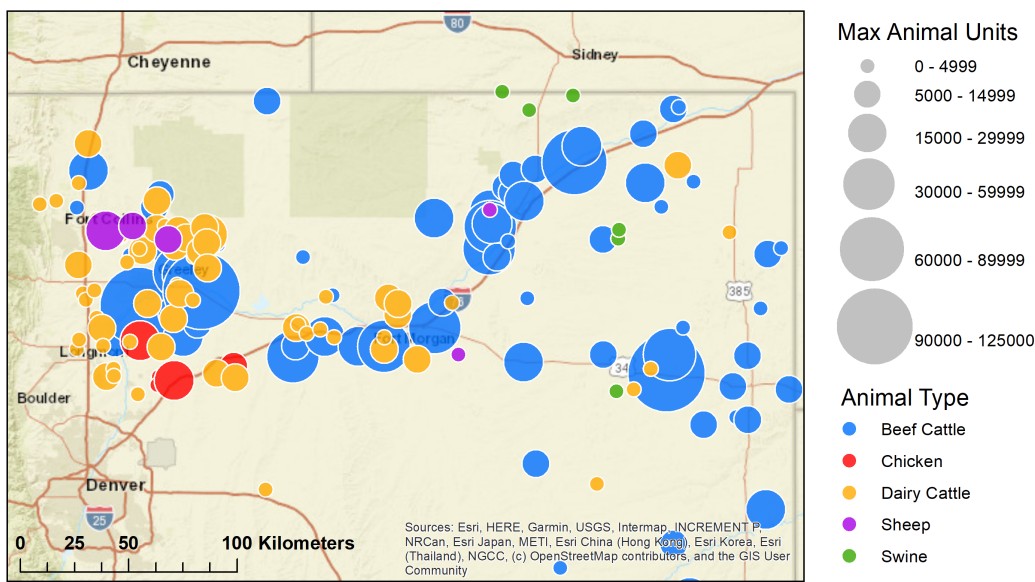

**Figure 1.** Map of Northeastern Colorado. CAFOs are colored by animal type and sized by max animal units. Animal units are equivalent to 1 live beef cattle such that 1 head of beef cattle = 0.7 dairy cattle= 2.5 swine = 10 sheep = 100 poultry (CDPHE, 2017). Note that the maximum animal units represent the maximum animal capacity of a given facility and not necessarily the actual number of animals present at that facility at the time of sampling. CAFO data as of 2017 registered with the Colorado Department of Public health & Environment (CDPHE, 2017).

The UWKA was instrumented to measure $CH_4$, $NH_3$, and $C_2H_6$ mixing ratios. A Picarro G2401-m flight-ready analyzer measured $CH_4$, carbon monoxide (CO), carbon dioxide ($CO_2$), and water vapor ($H_2O$) at 0.25 Hz through infrared cavity ring-down spectroscopy (Crosson, 2008). This Picarro and other Picarro models were tested previously and found to be stable and suitable for airborne field measurements (Richardson et al., 2012). Two separate Aerodyne commercial quantum-cascade

tunable infrared laser direct absorption spectrometers (QC-TILDAS) measured $NH_3$ and $C_2H_6$. These instruments are described in detail in Yacovitch et al. (2014), Pollack et al. (2019), and Pollack et al. (2022). $NH_3$ measurements are collected at 10 Hz and averaged to 1 Hz for reporting and this analysis; $C_2H_6$ measurements are collected and reported at 1Hz. All chemical data was adjusted for time lag between instruments and further averaged to 0.25 Hz for emission calculations. Instruments were calibrated on the ground before and after flights. Instrument zeros were routinely measured in flight by overblowing the

instrument inlets with a bottled source of synthetic "zero" air. Other in-situ measurements from the UWKA standard instrument package included pressure, temperature, three-dimension winds, GPS position, aircraft altitude, and heading. The wind speed is measured at 25 Hz, then averaged to 0.25 Hz. The precision on the reported wind speed is 0.14 m s-1 with an expected wind direction precision of 5 degrees (Strauss et al., 2015).

## 2.2 Flight Patterns

Flights were designed to identify the best flight patterns to simultaneously quantify $CH_4$ and $NH_3$ emission fluxes, $CH_4$ to $NH_3$ ratios, and $NH_3$ deposition downwind of CAFOs. Prior to each flight, forecast meteorology was used to identify the ideal CAFOs to sample based on prevailing wind direction, isolation from other CAFO plumes, and other logistical constraints (e.g. proximity to urban areas, towers and airports). Once airborne, the pilot conducted a vertical profile to characterize the mixed boundary layer (MBL). Selected CAFOs were located by coordinates and, once close enough, by sight. Once a selected CAFO was identified by sight, the UWKA pilot would perform a visual safety inspection of the area and then fly up/down in a spiral pattern centered on the selected CAFO. The aircraft proceeded to circle the target CAFO at a low altitude to confirm in situ enhancements of $CH_4$ and $NH_3$ mixing ratios and the direction of the outflow plume. Flight altitudes ranged from 0.1 km AGL to 3 km AGL, and near-CAFO flight altitudes depended on safety constraints.

$NH_3$ deposition calculations are not the focus of this work, however, observations of deposition require multiple downwind observations. $NH_3$:$CH_4$ ratios should be calculated near the source, as documented in Pollack et al. (2022) which used the same data set as this work. We investigated both spiral (which can be completed quickly) and horizontal transect flight patterns (which require much more flying time), shown in Fig. 2. The spiral patterns can be executed in quick succession, provide information as to the upwind background and provide multiple downwind distances for analysis of $NH_3$ deposition. However, the spiral transects were found to be undesirable for quantifying emissions because the aircraft could not get far enough outside the plume to characterize background conditions. For example, the further downwind spiral transects had high enhancements of $CH_4$ and $NH_3$ for the full width of the spiral, indicating that the UWKA did not leave the plume. The horizontal transects, while ideal for emission calculation, did not provide much information as to the evolution of downwind $NH_3$ deposition. In order to sample the full plume efficiently and provide multiple downwind observations, racetrack patterns or boxes were later identified as a preferred approach for future sampling.

## 2.3 Emission Calculations

CAFOs suited for emission calculations were identified based on the following requirements: (1) enhancements of $CH_4$ and $NH_3$ above background conditions, (2) flight path includes multiple transects downwind at different altitudes within the MBL, and (3) enough data near the CAFO, but outside the CAFO plume, to characterize background mixing ratios. Only one CAFO sampled on November 13, 2019 satisfied the stated requirements to be suitable for emission quantification and is used in the remainder of this work. The flight on November 13, 2019 (denoted as F2 from here forward) occurred during a period with strong and steady winds (average wind speed of 8.4 ) from the north-northeast (average wind direction 32°). The MBL was well mixed with a top at $1200 \pm 150$ m AGL (Fig. S1). The flight track was carefully planned to target a specific feedlot. The aircraft performed spiral transects at distances ranging from 4-14 km downwind of the feedlot, and stacked horizontal transects were conducted 12 km downwind. The horizontal transects were extended to account for high ammonia and methane values observed during the flight. However, it was found that this extended area included more than one feedlot, as determined by NOAA HYSPLIT back trajectories performed on the two peaks of the plume (Figure S1). The analysis showed that the entire

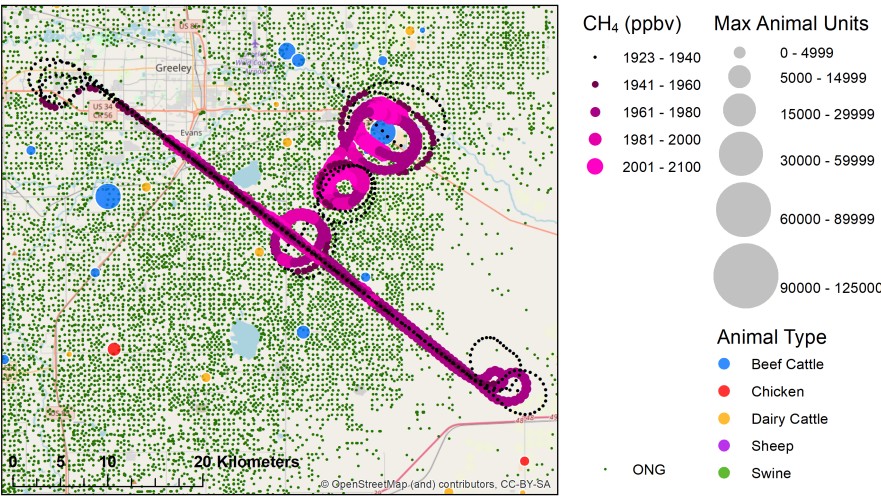

**Figure 2.** Observed CAFO flight path colored-filled with CH$_4$ (ppbv). Animal operations are indicated by the different colored circles as in Figure 1. Green dots represent ONG wells, data of ONG as of 2015. (Colorado Department of Natural Resources Oil & Gas Conservation Commission, 2016).

width of the plume contained emissions from multiple feedlots. The combined maximum capacity of all the feedlots within the plume was estimated to be 173,800 maximum heads based on the 2017 Colorado Department of Public health & Environment (Colorado Department of Natural Resources Oil & Gas Conservation Commission, 2016). The sampling region during F2

is surrounded by many ONG wells and is located 16 km southeast of Greeley, Colorado, an urban area with a population >100,000.

We used a mass-balance approach to calculate emissions. Aircraft mass-balance has been used to quantify emissions from a variety of source types (examples include Cambaliza et al. (2014); Caulton et al. (2014); Karion et al. (2015); Peischl et al. (2015, 2018)). Briefly, the mass flow rate of a species through a crosswind plane downwind of the source is approximated by

145 the integration of enhancement above a background concentration over the width and height of the plume. The emissions are derived using Eq. 1 shown here:

$$M(u) = \int\limits_{0}^{Z_{MBL}} \int\limits_{-x}^{x} (C_u - C_b) \times U_\perp \times \rho(z)\, dx\, dz \tag{1}$$

In Eq. 1, $M$ represents the molar flux (moles s$^{-1}$ ) of a gas downwind of the source. To find the enhancement, the local background concentration, $C_b$ (ppbv), is subtracted from the measured concentration, $C_u$. The ideal gas law is used to calculate

the air density ($\rho$) at every data point, using the universal gas constant at $\sim$8.31 J mol$^{-1}$ K$^{-1}$. The values $\pm x$ are the horizontal limits of the plume width from the center point, and $Z_{MBL}$ is the top of the MBL. The full plume's width was identified by consistent NH$_3$ values above 5 bppv. The plume's height is constrained to be from ground level to the top of the MBL. A

vertical profile near the source was used to identify the MBL level using $H_2O$ vapor and calculated potential temperature (Fig. S2). We used a constant value to represent the regional background. The background region was determined from the edges of the horizontal transects for the observed CAFO and used to calculate average values for $CH_4$, $NH_3$ and $C_2H_6$.

We transformed all observations surrounding the target CAFO sampled during F2 (Fig. 2) onto a polar coordinate system $(r,\theta)$ using the center of the target CAFO as the origin, following the process described in Nathan et al. (2015). The location of the data point on the polar coordinate system $(\theta)$ is perpendicular to the theoretical flux surface. So that, $U_\perp$ (m s$^{-1}$) is the corrected perpendicular wind by taking the cosine of the location of the data point on the polar coordinate system $(\theta)$ subtracted by wind direction $(\phi)$, multiplied by the wind magnitude $(V)$:

$$U_\perp = cos(\theta - \phi) \times V \tag{2}$$

## 2.4 Uncertainty Analysis

We conducted an uncertainty analysis using a Monte Carlo approach, creating a pseudo distribution of the data using observed means and standard deviations and recalculating the emissions rates. Due to the nature of the plume, not all variables may be represented using a pseudo distribution. We define the final emission uncertainty as the magnitude of the change in the emission from a combined function of the pseudo distributions of the following five parameters: background value, perpendicular wind speed, density, MBL depth, and instrumental uncertainty. The Monte Carlo recalculations were first done for individual parameters then as a combined pseudo distribution of perpendicular winds, density, background values, and MBL depth to calculate the final uncertainty. The uncertainty for the attribution methods followed the same approach but is addressed separately.

For the uncertainty analysis, we used a Gaussian distribution based on the mean and standard deviation of the original background value to select a new background value randomly. We found the density to have a pattern of increased values to one side of the horizontal transects. Therefore, to account for the pattern, we created a Gaussian distribution using the standard deviation found in each transect and added that onto a moving mean of 1 minute. The MBL height pseudo distribution was formed using a uniform randomly selected value of $2600 \pm 150$ m AMSL. Throughout the day, the MBL height changed by an average of $\sim$150 m. The perpendicular winds consist of two variables: wind speed and wind direction. For each separate transect, we created a pseudo-Gaussian distribution of wind speed and wind direction based on the transects' mean and standard deviations for each variable. These Gaussian distributions of wind speed and direction were recalculated through Eq. 2 to produce the new perpendicular wind speeds. The emission recalculations were done 1000 times using all four parameters for $CH_4$, $C_2H_6$, and $NH_3$. The uncertainty of final emission estimates were calculated by using the 95% confidence interval (CI) from the Monte Carlo approach divided by the average emission.

When considering the effect of instrumental uncertainty we only examine the factors actually impacting the measurements used in the mass-balance equation: the delta value ($C_u$-$C_b$). There are two factors that impact this value: (1) the intercept (bias), and (2) the slope (calibration factor) of the instrument. The bias, which can be affected by drift in the zero reading, is quantified for $NH_3$ and $C_2H_6$. To further clarify why these are the only instrumental uncertainty factors we evaluate and why total uncertainty does not affect our results we consider the following hypothetical situation: two otherwise identical

instruments are used to quantify the same enhancement. One instrument reads a background value of 1E6 +/- 0.1 and peak value of 1.000001E6. The second reads a background value of 1 +/ 0.1 and an enhancement of 2. The resulting delta values have the same absolute value and uncertainty because of how error propagates through subtraction regardless of which instrument was accurate. We assume that the variability in the bias is dwarfed by real variability in the background, or if the bias is actually large it similarly affects the background reading. Thus, we expect our analysis of background variability to be the appropriate metric to account for bias.

The uncertainty on the calibration factor is also possible to analyze and include. As far as we are aware, this source of uncertainty does not appear to be routinely reported or incorporated into uncertainty analysis in the mass balance literature. We assumed that the applied calibration factor can vary randomly and applied a randomly picked factor to the delta value to create a pseudo distribution of possible enhancements. Note that the accuracy of the sensor doesn't matter. If the uncertainty of the delta value is calculated in the fashion described, there would be equivalent uncertainty on an accurate or inaccurate reading provided the variability of the background and calibration factor was the same. The variability of the calibration factor was <0.1% for $CH_4$ and 9% for $C_2H_6$. For $NH_3$, Pollack et al. (2019) found the variability to be 2%.

The effects of instrumental precision, which affect both background values and enhancements are neglected because we average a large area to calculate the background and bin average 5 s of data for the enhancements. Averaging has the effect of decreasing the random error. The 1-Hz inflight precision is already low at 1 ppbv for $CH_4$ (Picarro), 200 pptv for $C_2H_6$, and 60 pptv for $NH_3$ (Pollack et al., 2019). Comparatively, the observed variability in the background values dwarfs the error from precision at 33 ppbv, 3 ppbv and 4 pbbv for $CH_4$, $C_2H_6$ and $NH_3$, respectively, thus we would expect the error introduced from precision to be negligible.

## 2.5 $CH_4$ Attribution

The two methods described here were applied to either all the data within the MBL during the flight (abbreviated F2), or only the downwind transects used for emission calculation (abbreviated Transect). This provides a total of four scenarios that will be analyzed. Ratios are reported in percentage (ratio x 100%) across all methods.

### 2.5.1 Subtraction Method

The first attribution approach, referred to as the subtraction method (SM), removes the $CH_4$ emissions related to ONG and attributes the excess $CH_4$ emission to the CAFOs. Rather than using inventory estimates for interfering sources, we calculated observation-based $C_2H_6$: $CH_4$ ratios. The F2 ratio (11 ± 0.02 %) is from Pollack et al. (2022). Briefly, $NH_3$ values >5 ppbv were used to screen out data points associated with CAFOS. The remaining data points were assumed to be associated with ONG sources, which is consistent with observations in this region that CAFOs and ONG are the dominant sources (Kille et al., 2019). The slope was calculated using least squares orthogonal distance regressions (ODR) and is equivalent to the $C_2H_6$: $CH_4$ enhancement ratio. Such a ratio can also be calculated from the CAFO transects as there was a region of $CH_4$ and $C_2H_6$ signal that did not include $NH_3$ interference located on the northwest end of the transect. The resulting ratio for the Transect data following the identical calculation is 14.7 ± 0.6%. The ONG $CH_4$ emissions were removed through Eq. 3:

$$M_{CH_{4_{ag}}} = M_{CH_4} - CH_4 : C_2H_6 \times M_{C_2H_6} \qquad (3)$$

The inverse of the $C_2H_6$: $CH_4$ ratio is used with the emission rate of $C_2H_6$ (moles s$^{-1}$ ) to retrieve the portion of $CH_4$ associated with ONG; that value is subtracted from the total $CH_4$ emission rate to result in the $CH_4$ emission associated with the CAFOs. Finally, the molar emission rate (moles s$^{-1}$ ) is multiplied by the molar mass to return the emission estimate in grams s$^{-1}$.

### 2.5.2   Multivariate Regression

An alternative to the SM is to directly calculate the CAFO $CH_4$ emissions from an $NH_3$: $CH_4$ ratio and the $NH_3$ emission rate using MVR. Because of the widespread ONG activity in this region, it is not always possible to have clear regions to calculate a ratio using a traditional regression approach (as in the SM). Indeed, there is also concern that the ratios calculated in the SM may not be accurate due to the influence of diffuse CAFO/agricultural signals. Fig. 3 presents $C_2H_6$ and $NH_3$ mixing ratios plotted against $CH_4$. In both plots there are elevated regions of the other species (i.e. regions of elevated $C_2H_6$ in the 230  $NH_3$ vs $CH_4$ plot). Instead, MVR using $CH_4$, $C_2H_6$, and $NH_3$ data can be used to calculate $NH_3$:$CH_4$ and $C_2H_6$:$CH_4$ ratios, as described in Pollack et al. (2022). Briefly, $C_2H_6$ and $NH_3$ are assumed to be independent tracers (associated with ONG and CAFOs, respectively) and $CH_4$ is the dependent variable, as shown in Eq. 4.

$$CH_4 = a + b \times NH_3 + c \times C_2H_6 \qquad (4)$$

In this equation, $a$ is the background $CH_4$ mixing ratio, $b$ is the inverse effective $NH_3$:$CH_4$ ratio and $c$ is the inverse SM 235  $C_2H_6$:$CH_4$ ratio. Unlike Kille et al. (2019) and Pollack et al. (2022), we did not subtract a background mixing ratios from the observed $CH_4$, $NH_3$ or $C_2H_6$ mixing ratios, thus the '$a$' variable actually represents the local background and we can compare its value to the observed background we identified at the edges of the transect. Kille et al. (2019) performed sensitivity analyses on their MVR results. Following the guidance of Kille et al. (2019), we only use the MVR results when all three variables are positive, $R^2 > 0.5$ and all variables are statistically different from 0. We also tested scenarios with background subtracted $CH_4$, 240  $NH_3$ and $C_2H_6$ to compare to the approaches used in Kille et al. (2019) and Pollack et al. (2022). We performed this analysis on the entire F2 dataset versus the Transect only data. The results of these sensitivity analyses are reported in Tables S1-2. The choice of background made no difference to the ratios, and only affected value a. This is consistent with Kille et al. (2019) and Pollack et al. (2022). Generally, scenarios where $NH_3$ and $C_2H_6$ were not background subtracted produced '$a$' values more consistent with the observed $CH_4$ background and were used for the remainder of the analysis. Slight differences in the MVR 245  results were observed from Pollack et al. (2022). This is attributed to the differences in the area used for MVR between these studies; Pollack et al. (2022) isolated specific source regions for MVR analyses while this study uses all of the data in the study region.

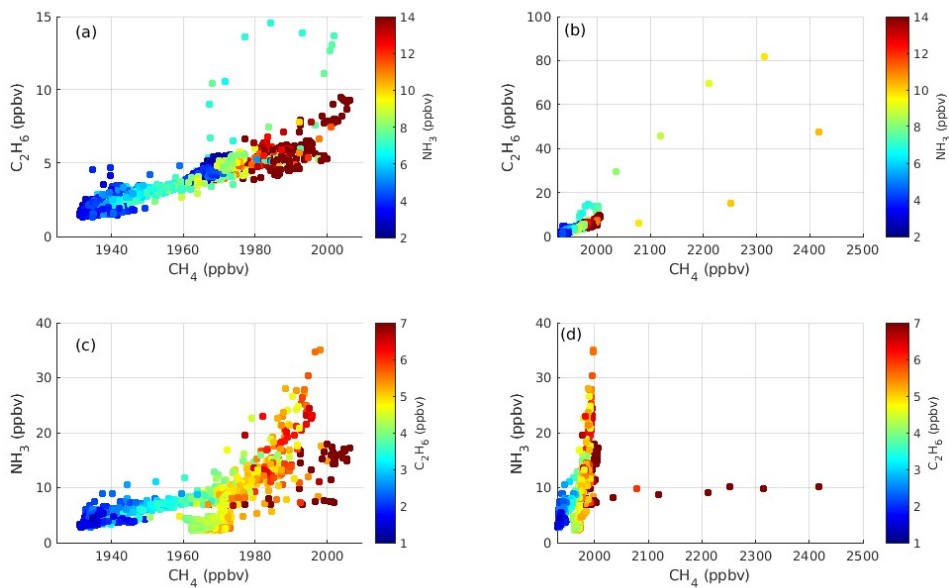

**Figure 3.** Scatter plots of (a) and (b) $C_2H_6$ versus $CH_4$ colored by the mixing ratio of $NH_3$ and (c) and (d) $NH_3$ versus $CH_4$ colored by the $C_2H_6$ mixing ratio sampled during the horizontal transects on 13 Nov 2019.

## 3  Results & Discussions

### 3.1  CAFO Emissions

Fig. 4 shows curtain plots of the measured $CH_4$, $NH_3$, and $C_2H_6$ mixing ratios from the horizontal transects. The curtain plots display the data from the flight paths in color-filled boxes representing the area used for the calculation in Eq. 1. In general, elevated $NH_3$ coincides with elevated $CH_4$, indicating the presence of an agricultural plume. However, a high $C_2H_6$ enhancement appears to be embedded in the agriculture plume, verifying mixed sources of $CH_4$ in this region. To isolate the agriculture plume and minimize the influence of other sources of $CH_4$, we created a mask around the $NH_3$ signal and screened

out points that were < 5 ppbv of $NH_3$ (Fig. 4e). Further, the plume was limited to regions with contiguous $NH_3$ values above 5 ppbv. This removed some isolated areas of $NH_3$ enhancement at the lowest altitude. Because the lower transects showed a consistent drop in $NH_3$, we extrapolated that boundary upward to the highest transect as indicated by the vertical dotted line on the left side of Figure 4. This removes some signal that is primarily downwind of Greeley, CO. The horizontal dotted line represents the boundary layer, and points above this layer were excluded. This threshold was identified by Pollack et al.

(2022), and was calculated as two times the maximum $NH_3$ mixing ratio observed outside of the plumes of individual CAFOs and within the MBL. The resulting mask was used to limit the area integrated for emission estimates according to Eq. 1. The calculated emission rates are 3330 g s$^{-1}$, 609 g s$^{-1}$ and 542 g s$^{-1}$ for $CH_4$, $NH_3$, and $C_2H_6$, respectively.

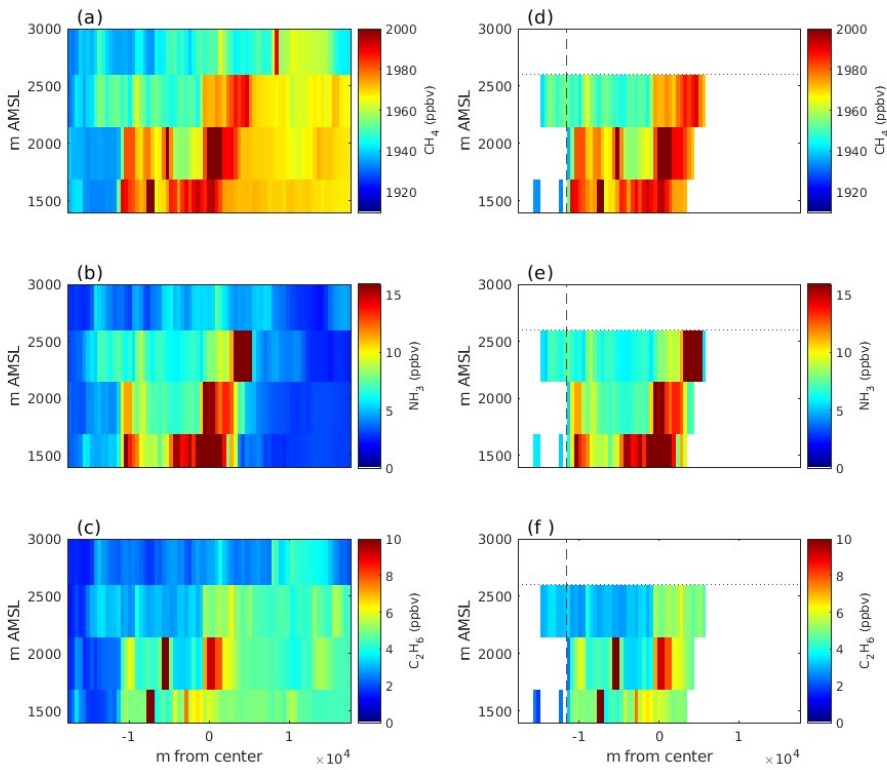

**Figure 4.** Vertical curtain plots of horizontal transects a) $CH_4$, b) $NH_3$ and c) $C_2H_6$ 12 km downwind of the observed CAFO. For reference, the surface is located at 1400 MSL. Vertical curtain plot of d) $CH_4$, e) $NH_3$ and f) $C_2H_6$ where $NH_3 > 5$ ppbv downwind of the observed CAFO. Points above the boundary layer were removed for calculation. Additionally, points on the left side of the curtain figure, above 5 $NH_3$ ppbv, were removed due to their likely origin from Greely, CO.

We present an in-depth look at the uncertainty contributed by each parameter for $CH_4$ emissions only. Perpendicular winds, density, and MBL depth are consistent in the calculations of $NH_3$ and $C_2H_6$ emissions; thus, we only discuss total uncertainty for these. Total uncertainty distribution plots are shown in Fig. S3.

Variability in density had little impact ( < 1%) on the final $CH_4$ emission rate. The winds during the plume had a standard deviation of ±1.3, ±1.9 and ±1.8 m/s for lowest to highest transect altitude respectively, with small changes in the wind direction with increased altitude. The wind speed was recalculated into the perpendicular wind and showed minimal change to the final $CH_4$ emissions by (∼4%). The location of the background in this study had a relatively consistent $CH_4$ mixing ratio ( mean = 1933 ppbv, standard deviation = 1 ppbv). The background is similar to the regional background (1990 ppbv) identified by Pollack et al. (2022). Background variation affects the final $CH_4$ emissions by 5%. $NH_3$ had higher variations in background values leading to increased uncertainty overall. MBL height was the largest driver of uncertainty and was associated with an

8% change in CH$_4$ emissions. The significant uncertainty due to MBL depth is expected; as changes in MBL depth would result in the interpolation of a different area without a response in concentration measurements. Thus, if the MBL depth would

have increased there is likely to be a corresponding decrease in concentrations. The absence of data between the lowest altitude transects and the surface may affect the accuracy of the results, but the associated uncertainty cannot be quantified because we do not have data at the surface. Instrumental uncertainty had little impact, changing the emission rates by <1% for all, CH$_4$ (0.3%), NH$_3$ (0.6 %), and C$_2$H$_6$ (0.5%).

Cambaliza et al. (2014) carried out a detailed analysis on uncertainty from aircraft mass balance calculations. They found that

MBL depth and background mixing ratios may have uncertainties up to 19% and 21%, respectively, on the overall emission rate estimate. Other studies have confirmed that uncertainty associated with the MBL depth makes a large contribution to overall uncertainty for these types of calculations (Karion et al., 2013; Peischl et al., 2015). The winds in the mass balance are assumed consistent from the release of the emissions to location of measurement. Therefore, the natural variability of winds may contribute to the uncertainty more with less consistent winds (including direction and speed) and may have a large effect

on the final uncertainty similar to MBL and background value (Karion et al., 2013; Peischl et al., 2015). During F2 winds were consistent and made a smaller contribution to the overall uncertainty on this particular day than noted by these other studies.

The total uncertainty for CH$_4$ emissions calculated from horizontal transects was $\pm$ 10%. For the emissions estimates for C$_2$H$_6$ and NH$_3$, the total uncertainty was $\pm$ 14% and $\pm$ 17%, respectively. Other factors that may influence the accuracy of the emissions include smaller-scale variations in the mixing ratios and regions of the plume the flights did not sample. As it

is not possible to sample the entire MBL from top to bottom, the vertical spacing between the horizontal transects may result in errors. Errors associated with interpolation were not explored here, but are expected to be small (Cambaliza et al., 2014). Further, when the bulk of the mixing ratio is near the ground is seen have the most uncerntainy (Gordon et al., 2015). The elevated mixing ratio values of CH$_4$, NH$_3$ and C$_2$H$_6$ were seen throughout the MBL. The uncertainty estimates are specific to this flight. Different meteorological conditions can produce different uncertainty estimates. In particular, MBL uncertainty or

growth may significantly affect the magnitude of the uncertainty. The uncertainty analysis methodology presented here can be applied to any future mass-balance emission estimates.

## 3.2 CAFO CH$_4$ Attribution: Comparison of Methods

Results from the SM and MVR CAFO CH$_4$ attribution and associated uncertainty are presented in Table 1. Reported uncertainties in Table 1 represent results from combining the ratio uncertainty to the rest of the pseudo distributions as described

in Sec 2.4. Both the magnitude of the attributed CAFO CH$_4$ emission and its uncertainty vary between approaches. These methods represent typical approaches that can be undertaken to isolate the emissions for a given facility. In order to identify the optimal method, we identify four criteria: (1) the relationships should predict the total CH$_4$ well; in other words traditional goodness of fit (GOF) values should be optimal (the average residual of the fit should be near 0 and R$^2$ values should be high)., (2) the relationships for NH$_3$:CH$_4$ and C$_2$H$_6$:CH$_4$ should be consistent with observations, (3) the method should err on the side

of being conservative, meaning it should be more likely to under-attribute than over-attribute the CAFO emissions, and (4) uncertainty of the result should be low. As currently presented these criteria are qualitative, because each analysis is unique.

**Table 1.** CH$_4$ Attribution Sensitivity

| Approach | Data | NH$_3$:CH$_4$ (%) (SE)[a] | C$_2$H$_6$:CH$_4$ (%) (SE)[a] | CAFO CH$_4$ (g s$^{-1}$) (95% CI) | CAFO Relative Uncertainty (%) | n | Comments[c] |
|----------|------|----------|----------|----------|----------|----------|----------|
| SM | F2 | 87[b] | 11 (0.02)[b] | 697(±423) | 61% | 12,195[b] | Fails criteria 1,2,3 & 4 |
| SM | Transect | 45 | 14.7 (0.7) | 1,359 (±442) | 34% | 201 | Fails criteria 1,2 & 3 |
| MVR | F2 | 157 (2) | 15.8 (0.1) | 366 (±60) | 17% | 6,715 | Fails criteria 1 & 2 |
| MVR | Transect | 92 (6) | 17 (0.3) | 626 (±122) | 20% | 1,568 | Passes all criteria |

[a] SE = standard error. [b] Pollack et al. (2022). [c] Criteria are defined in Sect. 3.2.

We will discuss the implications and further refinements of these criteria later on. These criteria are presented in order of their importance to the final recommendation. This approach ensures that low uncertainty is not the primary deciding factor.

To investigate criteria 1, we calculated total fits, residuals and R$^2$ values for all four methods and assessed how well the
NH$_3$:CH$_4$ and C$_2$H$_6$:CH$_4$ ratios represented the transect data. The central assumption is that regardless of attribution method, all methods should predict the observed CH$_4$ values well as evaluated by traditional metrics of goodness of fit (e.g. R$^2$, residuals). MVR directly produces both ratios such that predicted CH$_4$ can be calculated from the NH$_3$ and C$_2$H$_6$ time series. To construct a CH$_4$ prediction for the SM, which does not require calculation of NH$_3$:CH$_4$ ratios, we inferred the effective NH$_3$:CH$_4$ ratio from the ratio of the calculated NH$_3$ emissions to the CH$_4$ attributed to the CAFO by SM. The resulting ratio is then used in
Eq. 4 to produce an effective multivariate regression prediction. Table S3 shows the full range of variables for each scenario. GOF statistics are calculated for the transect data only.

The results of this comparison are shown in Fig. 5, with the prediction limited to the Transect data, because this is the data that is actually used to calculate the CH$_4$ emission rate. The Transect MVR approach was the method with the best GOF statistics including an average residual closest to 0 and the highest R$^2$ value and the only approach that passes criteria 1. In
some ways this is unsurprising because fits are calculated by minimizing the sum of the residual squared, and this is the only method that directly fits the Transect data (Skoog et al., 2004). Still, this analysis does provide the means to compare the other methods and can be useful when site specific MVR is not possible. The background values were consistent in all methods and consistent with observations. The three other methods all overpredicted CH$_4$ and had average negative residuals and lower R$^2$ values. The over-prediction suggests that these methods produce an incorrect relationship between CH$_4$ and one (or both) of
the tracer gases, but it is not possible to identify which relationship is incorrect. The method that best simulates the data after the Transect MVR is the F2 MVR.

The effective NH$_3$:CH$_4$ ratios produced for this method can also be compared directly in support of criteria 2. Criteria 2 is difficult to evaluate because ideally the actual ratios would be known. We can compare to literature values, compare values between scenarios, and look at the underlying assumptions of the calculations to evaluate this criteria. The C$_2$H$_6$:CH$_4$ ratios
calculated for SM and MVR varied by 40%, which substantially affects how much CH$_4$ is attributed to ONG. However, the

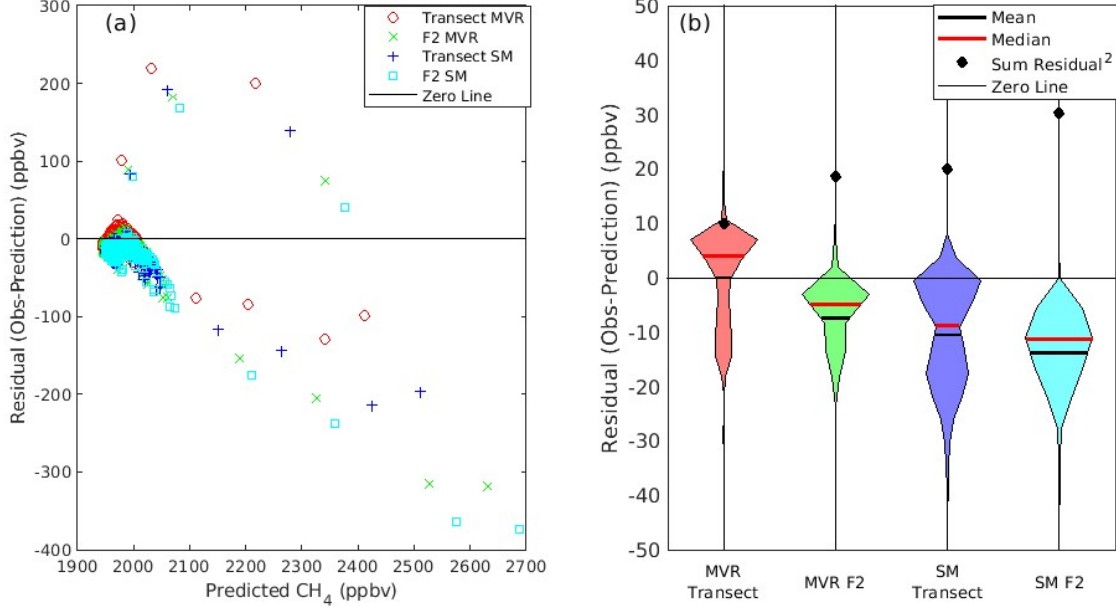

**Figure 5.** a) Residual $CH_4$ (observation - prediction) vs. predicted $CH_4$ (ppbv). The ratios corresponding to the transect MVR results are shown in red, the F2 MVR results are shown in green, the transect SM results are shown in blue, and the F2 SM results are shown in cyan. The black line in both plots represents the zero line. b) Violin plots of residuals for the four different methods. The black lines are the calculated mean residuals, the red lines are the median residuals and the closed black dots are the sum of the residuals squared (normalized by the minimum value x 10). The violin plot is ordered by increasing absolute average residual.

range of ONG ratios is broadly consistent with other observations in this region (Kille et al., 2017, 2019; Yacovitch et al., 2017; Peischl et al., 2018).

As a plume moves downwind from its source it is expected that the plume will disperse over the MBL. $NH_3$ has a short lifetime ($\sim$hours) and may be removed through dry deposition and transition into the particle phase even at relatively short dis-
tances downwind from sources Staebler et al. (2009); Miller et al. (2015). Thus, there may be lower concentrations downwind of gaseous $NH_3$ than expected from dilution alone. The range of reported $NH_3$:$CH_4$ ratios in the literature is quite variable and includes observations from 1-200% Eilerman et al. (2016); Golston et al. (2020). We used Pollack et al. (2022) as our primary reference for comparison. Comparison to this study revealed that the transects located $\sim$12 km downwind of the observed target CAFO do not have the same $NH_3$:$CH_4$ ratio as transects collected closer to the CAFO. There is also generally a wide
variety of ratios reported in Pollack et al. (2022) ranging from 80% to 270% across different facilities which illustrate why the MVR results from the full F2 data may not be appropriate to represent an individual site. In general, the SM produced ratios that were lower than MVR and observations near the source (Pollack et al., 2022). The SM Transect ratio is particularly low

(45%), about half the value of any other ratio produced from the other attribution methods or results from Pollack et al. (2022). This indicates that this approach actually fails criteria 2.

On a theoretical basis, the MVR ratios calculated from the F2 data include locations near other CAFOs and locations near ONG sources. Therefore, the F2 MVR results may not accurately represent the area around this particular plume. The SM using F2 data has similar challenges. The SM assumes that $C_2H_6$:$CH_4$ is constant throughout the area and that there are no other sources of $CH_4$ other than the CAFO and ONG. However, we know there may be other trace amounts of $CH_4$ from vehicles, waste areas, and other small sources. Generally, we would expect data nearest the source to produce the most accurate ratios
and pass criteria 2. This would generally exclude the F2 results and for this reason, we assume these results fail criteria 2.

    For criteria 3, the two methods are different in approach and it is quite easy to identify which approach is more conservative. The SM attributes anything outside of ONG to the CAFOs plume, thus there is no unattributed $CH_4$ for these methods. The SM is, therefore, expected to be less conservative and an upper limit of the CAFOs $CH_4$ emissions due to the possibility of other sources in the region.

In the MVR approach, the $R^2$ value of the fit gives an estimate of how much of the $CH_4$ signal is explained by the chosen tracers. The MVR fits had $R^2$ values of 0.72 for F2 and 0.74 for Transects, which leaves an amount of $CH_4$ that is not well correlated to $NH_3$ or $C_2H_6$. In this scenario, this excess $CH_4$ is left unattributed in the MVR approach (56% for F2 and 24% for Transect). A visual representation of the unattributed portion of the $CH_4$ signal for the transect MVR is presented in Fig. S4. The unattributed $CH_4$ is broadly distributed and cannot be attributed to any particular source. The MVR approach avoids
over-attributing the $CH_4$ signal to the CAFOs.

    In general, we expect the Transect MVR approach which uses the data closest to the source to provide the best estimate of these CAFOs emissions in accord with the first three criteria. This approach produces local ratios on par with the literature, has the closest average residual to 0, and is conservative in its approach to attribution. Transect MVR has slightly greater uncertainty than F2 MVR and both have lower uncertainty than SM because they do not require subtraction. The impact of
subtraction on uncertainty is evident in Table 1 and is the basis for including criteria 4. While the ONG ratios calculated either by SM or MVR using the F2 data differ by ~40%, the uncertainty on the results using MVR attribution is a factor of 3 lower. This is despite the fact that the SM ratio uncertainty is actually a factor of 5 lower than the MVR ratio. If we had used a comparable error on the $C_2H_6$:$CH_4$ ratio to MVR with SM, the error would approach 100%. The primary reason for the difference in uncertainty between the SM and MVR approaches relates to how error is propagated through subtraction vs.
multiplication or division. For subtraction, absolute errors add in quadrature, while for multiplication/division, relative errors add in quadrature (Skoog et al., 2004). The net effect is that error on the SM will be very high when the absolute amount to be subtracted is large. The ONG ratio is not directly used for attribution in the MVR approach, which is why the CAFO attributed emissions by the F2 MVR and SM approaches vary by ~30%. However, the relative uncertainty (standard error/ratio) on the $NH_3$:$CH_4$ MVR ratio is comparable to the relative uncertainty of the MVR ONG ratio.

The subtraction method is very similar to inventory subtraction in principle. Error estimates are often not provided for inventory data, but the error introduced to emission estimates by subtracting inventories may be substantial. Zimmerle et al. (2022) recently showed how ONG sources could vary and produce uncertainties in excess of estimates relying on traditional

inventory data. The MVR approach is generally attractive to avoid such errors when interfering signals are high and there is enough data to produce robust MVR results. The plume from the observed CAFOs was broad, allowing for multiple data points within the plume and produced results with low uncertainty (20 %). For sites with narrower and lower enhancements local MVR may not be possible. Our criteria would suggest that using MVR on the full flight data may be an appropriate proxy when local ratios with MVR cannot be calculated. However, the results from the MVR Transect and F2 data are statistically different in this case. Additional analysis may need to be done to ensure the MVR results from larger data sets are applicable to the site of interest. Repeat measurements or ground observations would help to constrain such results.

## 3.3  Additional Case Studies

To more thoroughly test the methodology we have so far developed for methane attribution in complicated regions, we have applied our strategy to a case study from a larger dataset of observations in the region. This data was collected as part of the Transport and Transformation of Ammonia (Trans2Am) campaign during Summer '21 and '22, where updated flight plans were performed for CAFOs in the same region as this work. While full results from that campaign are outside the scope of this manuscript, we compare to one research flight (research flight 13, denoted as RF13 in the remaining text) in the same vicinity as the facilities measured during the original F2. The updated flight plans, shown in Figure 6, included multiple downwind transects to allow computation of emissions at consecutive distances over the course of a ~1.5 hours. We calculated $CH_4$ and $NH_3$ emission rates and uncertainties identically to the previously described methodology. We would expect the $CH_4$ emission rate to be conserved at different downwind distances. The $NH_3$ emission may be affected by deposition to the surface and thus may remain the same or decrease downwind from a facility.

Table 2 shows results from RF 13, which was collected on August 23, 2021 near Greeley, CO. The target CAFO was the same target feedlot in F2. The top of the MBL was located at $2000 \pm 200$ m AGL. An average temperature of 24 °C and southwesterly winds with an average wind speed of 4.1 m s$^{-1}$ were observed during the flight and a total of five downwind transects were performed. Background values and uncertainties for $CH_4$ were obtained from the MVR fit and varied for each transect, ranging from 1886 to 1927 ppbv. The background $NH_3$ value (3.023 ppbv) was calculated by constraining observations to those associated with the target CAFO (Fig. 6) and identifying the minimum $NH_3$ concentration. Enhanced concentrations of $CH_4$ and $NH_3$ were observed during all transects, spanning about 17 km downwind of the target facility. CAFO $CH_4$ was attributed using the transect MVR approach for all five transects, described previously.

Attributed $CH_4$ ranged between $189 \pm 55$ g s$^{-1}$ to $533 \pm 162$ g s$^{-1}$ with a total average of $365 \pm 89$ g s$^{-1}$. The average emission rates for this site convert to $12 \pm 3$ g $CH_4$ head$^{-1}$ hr$^{-1}$ and $13 \pm 3$ g $NH_3$ head$^{-1}$ hr$^{-1}$. There is a decreasing trend in $NH_3$:$CH_4$ ratio observed as the transect distance increases indicating $NH_3$ depletion is occuring, likely by dry deposition. The per head values use the combined maximum number of cattle from all facilities within the plume (109,500 cattle), according to the CAFO Permit Database 2021 (CDPHE, 2022). The total average uncertainty range (95% CI) for this site was calculated from the distribution of the sum of all 5 transect monte carlo simulations. The total average relative uncertainty for this case (24%) is higher than the uncertainty of the previous case (20%) due to different environmental conditions. While there is considerable variability in the $CH_4$ emissions from this site, we note that the $NH_3$ emissions show similar variability. There is a factor of

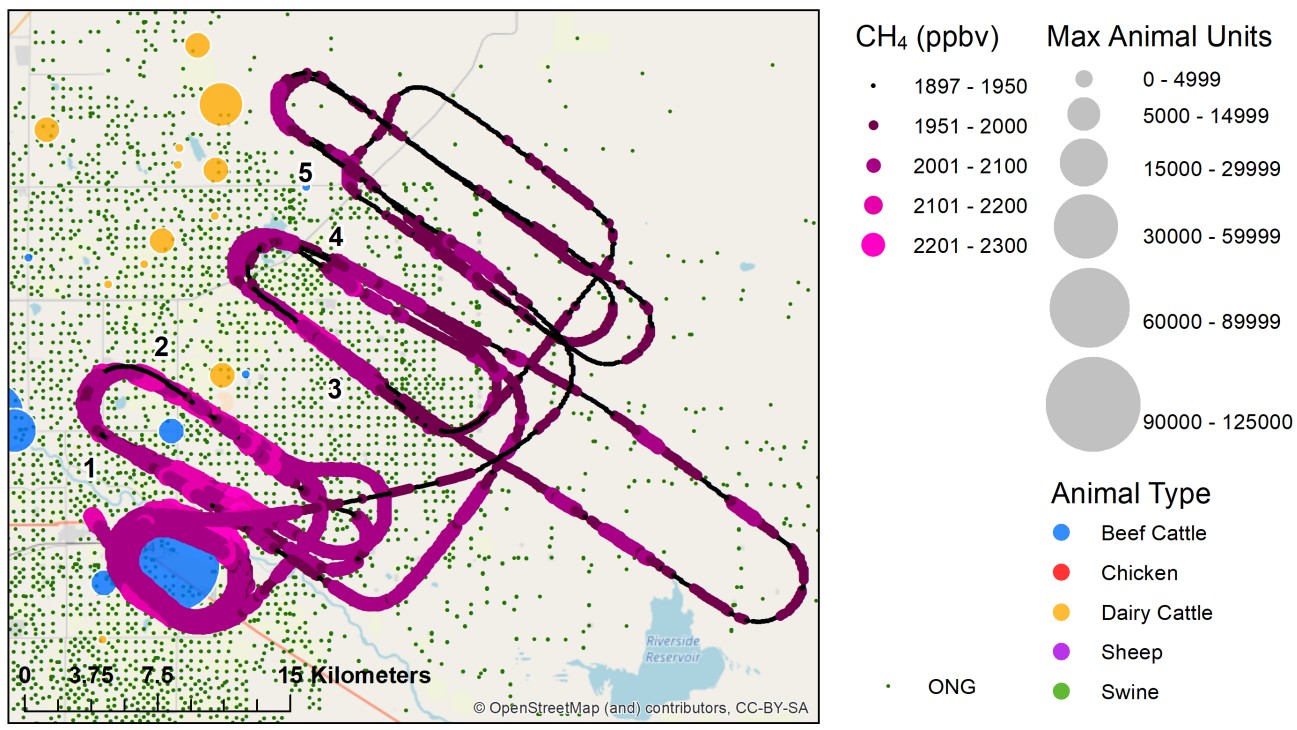

**Figure 6.** Path of TRANS2Am RF 13, colored by $CH_4$ (ppbv). Black numbers represent the corresponding transect number. Animal operations are indicated by the different colored circles as in Figure 1. Green dots represent ONG wells, data of ONG as of 2015 (Colorado Department of Natural Resources Oil & Gas Conservation Commission, 2016).

**Table 2.** $CH_4$ Attribution for Trans2Am RF13.

| Transect Number | Distance from Feedlot (km) | NH$_3$:CH$_4$ (%) (SE)[a] | C$_2$H$_6$:CH$_4$ (%) (SE)[a] | CAFO CH$_4$ (g s$^{-1}$) (95% CI) | CAFO Relative Uncertainty (%) | CAFO NH$_3$ Uncertainty | NH$_3$ Relative Uncertainty |
|---|---|---|---|---|---|---|---|
| 1 | 2.8 | 152 (5) | 9 (0.6) | 189 ($\pm$55) | 29% | 307 ($\pm$86) | 28% |
| 2 | 5.5 | 128 (5) | 7 (0.4) | 482 ($\pm$99) | 34% | 656 ($\pm$127) | 19% |
| 3 | 11.1 | 95 (2) | 6 (3) | 297 ($\pm$91) | 17% | 300 ($\pm$91) | 30% |
| 4 | 12.9 | 83 (3) | 10 (0.7) | 533 ($\pm$162) | 20% | 469 ($\pm$139) | 30% |
| 5 | 12.9 | 87 (4) | 8 (0.5) | 322 ($\pm$70) | 22% | 270 ($\pm$60) | 30% |
| Average | 9.8 | 125 (2) | 8 (0.2) | 362 ($\pm$89) | 24% | 386 ($\pm$99) | 24% |

[a] SE = standard error.

2.7 between the minimum and maximum $CH_4$ emission observations and a factor of 2.4 between the minimum and maximum $NH_3$ emission observations. As the $NH_3$ emission rate is not affected by attribution method, this suggests that the variability in $CH_4$ is primarily caused by on-site or environmental conditions, not the MVR attribution method. Comparisons between the MVR and SM attribution methods for this facility show (Table S4) that the SM method produces more variable results (factor of 3.9 min-max range) and overpredicts the total average methane by about 49%, while the total average relative uncertainty associated with the SM method for this site is almost double the uncertainty associated with the MVR method. These results are in agreement with our previous results. The average per head $CH_4$ and $NH_3$ results for this site are not statistically different from the Nov '19 results despite the different wind direction, background values, etc.

## 3.4 Comparison to literature

A few recent studies have reported $CH_4$ emission rates from CAFOs. The emissions factors calculated from the Transect MVR method in this work are 13 ($\pm3$) g $CH_4$ head$^{-1}$ hr$^{-1}$ and 13 ($\pm2$) g $NH_3$ head$^{-1}$ hr$^{-1}$ in November 2019 (F2) and in 12 ($\pm$ 3) g $CH_4$ head$^{-1}$ hr$^{-1}$ and 13 ($\pm$ 3) g NH3 head$^{-1}$ hr$^{-1}$ August 2021(RF13). The calculation of emission rates as a value per head is based on the maximum amount of cattle allowed in the CAFOs, although the actual number of cattle in each CAFO at the time of the measurement is unknown. Our maximum number of cattle identified for the F2 data consisted of mostly beef cattle at 161,500 heads, and only 12,300 heads were dairy cattle (total 173,800 heads). The EPA estimated $CH_4$ emissions for beef cattle is $\sim$7.2 g head$^{-1}$ hr$^{-1}$ and previous studies range from 7 - 9 (EPA Annex, 2021; Golston et al., 2020; Hacker et al., 2016). Dairy cattle have high rates of $CH_4$ emissions due to higher exertion on the animal with rates ranging from 14 - 39 g head$^{-1}$ hr$^{-1}$ and the EPA national average is 18 g head$^{-1}$ hr$^{-1}$ (EPA Annex, 2021; Leytem et al., 2011; Bjorneberg et al., 2009; Griffith et al., 2008). Golston et al. (2020) conducted a ground-based study in NCFR and found the highest emission rates in dairy cattle at 39.32 $\pm$ 3.92 g $CH_4$ head$^{-1}$ hr$^{-1}$. In contrast, their reported emissions from beef cattle were much lower and similar to other studies (9.48 $\pm$ 0.93 g $CH_4$ head$^{-1}$ hr$^{-1}$). However, they noted there was a significant difference in emissions rates of repeat observation for an individual CAFO.

The difference between the $CH_4$ emissions per head of cattle in this work compared to prior works could be due to the management of the CAFOs. Enteric fermentation can be altered through the composition of the food given to the animals. For instance, the type and maturity of diets provided to the animals may modify the nutrients and digestibility of the food (Archimède et al., 2011). $CH_4$ emissions from cattle are also known to depend on exertion on the animals including exercise and stress (EPA, 2022). In this study, the food source and feeding schedule were unknown. We also note that one CAFO was extremely large with 98,000-100,000 reported maximum cattle heads during the time of study and one of the largest in Colorado. Most emissions estimates are based on considerably smaller CAFOs.

The $NH_3$ emissions per head reported here are also slightly than previous estimates, which ranged from 2-12 g head$^{-1}$ hr$^{-1}$ (Staebler et al., 2009; Golston et al., 2020; Hacker et al., 2016). The $NH_3$ emission per head calculation in our work does not require removal of interfering signals like the $CH_4$ emissions. The $NH_3$ emissions are highly affected by meteorological conditions including seasonality and time of day. Waste that is stored outside may exacerbate $NH_3$ emissions in warmer temperatures (Montes et al., 2013). There is also evidence of a diurnal pattern of $NH_3$ emissions from CAFOs with a peak

during noon local time, near the time of sampling in this work (Shonkwiler and Ham, 2017). $NH_3:CH_4$ ratios follow a similar pattern with higher ratios during the midday and lower ratios at night (Eilerman et al., 2016).

## 4 Conclusions

We demonstrate an approach to isolate and quantify $CH_4$ plumes of agriculture CAFO sources with interfering sources by using SM and MVR methods. The SM method uses $NH_3$ as a tracer to identify a CAFO plume before using a $C_2H_6:CH_4$ ratio to remove any ONG $CH_4$ emissions interfering with the CAFO source. It is an optimal method when there is little ONG influence to keep the error introduced from subtraction small. The MVR method uses $C_2H_6$ and $NH_3$ as tracers and provides lower uncertainty on emission estimates. This approach is appropriate when tracer data is available and there is enough signal to produce statistically significant relationships and thus site specific ratios. The criteria to identify the best approach may be useful for isolating and attributing emissions from specific sources in other regions. Overall, our best estimates of emissions from the observed CAFOs are 13 ($\pm$2) g hr$^{-1}$ head$^{-1}$ for $CH_4$ and 13 ($\pm$2) g hr$^{-1}$ head$^{-1}$ for $NH_3$. These findings are significantly higher than EPA inventory estimates and previous studies highlighting the need for more observations and estimates of CAFO $CH_4$ and $NH_3$ emission rates.

*Data availability.* Merged data files for the F2 data will be made available from WyoScholar https://wyoscholar.uwyo.edu/ This long-term repository is open-access and provides freely available data stewarded by the University of Wyoming Libraries. The RF13 data collected as part of the TRANS2Am campaign is publicly available via the National Center for Atmospheric Research Earth Observing Laboratory: https://data.eol.ucar.edu/master$_lists/generated/trans2am/$

*Author contributions.* CRediT: https://authorservices.wiley.com/author-resources/Journal-Authors/open-access/credit.html

MEM: Writing-Original Draft, Formal Analysis, Visualization, Data Acquisition and Curation

IBP: Writing-Review & Editing, Data Acquisition and Curation, Funding Acquisition

EVF: Writing-Review & Editing, Funding Acquisition

KMS: Formal Analysis, Data Acquisition and Curation, Writing-Review & Editing

DRC: Conceptualization, Supervision, Writing-Review & Editing, Funding Acquisition, Data Collection

*Competing interests.* The authors declare no competing financial interests.

*Acknowledgements.* We would like to thank the University of Wyoming King Air facility for their support of this project. We would also like to thank Kristen Pozsonyi for help with instrument installation and data collection. This work was supported in part by the National Science Foundation (grant #2020151).

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
