# Peer review of "Technical note: Isolating methane emissions from animal feeding operations in an interfering location"

_EGUsphere, 2022_

## Referee Comment (RC1)

Review of "Technical note: Isolating methane emissions from animal feeding operations in an interfering location" by McCabe et al.

December 5, 2022

Reviewer Recommendation:

This paper is suitable for publication in ACP in structure and content but requires revisions to clarify key scientific conclusions.

Summary:

This method paper describes two approaches to isolate CH4 emissions in the NCFR where both ONG and feedlots contribute to the total emissions. The data is obtained from airborne in-situ measurements. CH4, NH3, and C2H6, with NH3 and C2H6 used as tracers for feedlots and ONG. The two methods that are compared for one CAFO from one research flight are a subtraction method and a multivariate linear regression method. MVR has lower uncertainty and fulfills the determined selection criteria in determining a CH4 emission from the feedlot. The final emission results of both CH4 and NH3 are higher than previous studies and inventory estimates and there is spread especially in the NH3:CH4 ratio compared to previous studies. The flight pattern during F2 in Nov 2019 was not the optimal pattern determined for the remainder of the TRANS2AM campaign.

**Specific Major Comments:**

1) Concern about chosen data set:
The publication by Pollack et al. (2022) utilized the exact same flight (F2) and performed similar analyses as reported in this manuscript such as the MVR and comparison of enhancement ratios. Pollack et al. (2022) also reported that the optimal flight track pattern would be racetrack or box patterns, whereas F2 used an approach that was determined to be less ideal (Lines 110-112, 113-115f of this manuscript). It is my understanding that TRANS2AM flights took place in 2021 and 2022 with such racetrack patterns flown (observed in UCAR field catalog). If this understanding is correct, I would suggest that flight data with a more optimized sampling strategy should be used as the basis for this technical note or an analysis of the errors resulting from non-ideal sampling should be included. My concern is that the results are not transferrable to analyses where the background of mixing ratios is more carefully determined from the racetrack patterns, which will cause unquantified biases, see comment 2) below.

2) Concern about secondary sources:
You consider the background of the mixing ratios to be constant and use the transect that is 12 km downwind of the targeted CAFO to support that conclusion. This assumption, if incorrect, is substantially problematic. One way this assumption could be invalid is if you observe 2 plumes. In my understanding, the ideal flight pattern would be racetrack or boxes where you fully capture any upwind influence along a full transect that is transported towards the horizontal transect downwind, which would capture any secondary plume passing through

your region of study. It appears to me looking at Figure 4 that you may have indeed measured two distinct sources contributing peaks in NH3 (and CH4), one near -10000m from center of the F2 transect and one that is likely the targeted CAFO at +/- 5000m from the center of the transect. Additionally, C2H6 (and CH4) is enhanced from -10000 to the far end of the transect (on the right of the plot) but is not enhanced at the end of the transect towards the left (<-10000m from center). Your conclusion in Line 384 is that the emissions per head from this study are higher than previous estimates. If there was another CAFO contributing to the second, smaller peak in Figure 4, then the per head emissions are higher than the actual per head emissions. Without this more ideal sampling strategy, how can you support your conclusion that you are observing an isolated source?

3) Concern about constraint for CH4 attribution:
In Line 199 is stated that F2 is all data within the MBL during the flight – but your flight started and ended in Laramie, Wyoming (see Line 83), which is in a very different area than the observed CAFOs, separated from the NCFR by a mountain range. All flight data that is outside of a certain boundary of the NCFR should be excluded instead of only those outside a certain altitude range.

**Minor Comments:**

Line 4: I suggest using present tense instead of past for "relied" unless the current method is something else. Then current method should be stated as well.

Line 20: It is unclear whether the 8.4% growth is for the US or globe or a specific region within the US. Assuming it is the US based on second half of the sentence, I suggest "grew 8.4% in the US"

Line 22f: Remove second mention of EPA reference

Line 64f: This sentence is somewhat true but also misleading and weakening your conclusion in that MVR is the optimal method. I believe you are trying to say that using more variables in the MVR for additional sources can falsify the results and is not appropriate. On the other hand, using fewer variables in the MVR and only those you are certain that are sources will not falsify the results for those known sources – additional possible sources will be treated in the extra term or background. I would suggest adding to the sentence first a quick focus on how MVR is appropriate, then what situations make MVR inappropriate.

Line 66: Missing an "a" or "the" prior to "methodology"

Line 78: Missing a space in "Greeley,CO"

Line 86: "COtextsubscript2" should be "CO2"

Line 87: Suggest "This Picarro and other Picarro models" instead of "This model and other Picarro models" to avoid confusion with atmospheric model

Figure 1: I suggest adding numbers for the middle two circles as well, as 1 and 125000 are the two extremes and it is nearly impossible to envision what capacity the intermediate circles represent.

Figure 1 caption: Are animal units determined by yourself or is there a reference? Are the equivalents with respect to CH4 or NH3 emissions or are the NH3:CH4 ratios between different animals identical? Please clarify.

Figure 1 caption: I suggest shortening the second sentence in the caption to "CAFOs are colored by animal type and sized by max animal units" as the color info is found in the legend.

Line 125: Do you mean 6.5° variability in the wind direction? Which instrument was used for wind and what is its measurement uncertainty (0.65° as currently stated seems too low in my experience)?

Line 126: "at t 4 km- 14 km downwind" should be "at 4-14 km downwind"

Line 191: Remove comma from "(2019),"

Line 193, 196: Once you use ppb, once you use ppbv. Pick one and check throughout paper and figures for consistent use of units.

Line 201: "(ratio x 100)" should be "(ratio x 100%)"

Line 228: "performed sensitivity analysis" should be "performed sensitivity analys**e**s"

Line 237: "regions for MVR analysis" should be "regions for MVR analys**e**s"

Line 238: I suggest moving Eq. 4 behind "as shown in Eq. 4." in Line 224 where it is described in the following sentence. Currently it seems to be floating without context prior to the next Section.

Line 246 and Figure 4: In Line 246 you state that you "screened out points that were <5 ppbv of NH3" but in Figure 4 I certainly seem to see cyan colors further to the left in the layer between 1500-2000 m AMSL as well as along the entire transect in the layer above 2500 m AMSL. The color bar indicates 5 ppbv is a darker shade than the cyan color.

Figure 4: How were the vertical curtain bins determined? Why does the curtain not start at 1500 m AMSL, since you stated in Line 81 the lowest flight altitude was 100 m AGL and state in the caption that the surface is at 1400 MSL?

Table 1: Currently missing the CI superscript in the table header

Line 272: Wind speed of 8.4 +/- 2.7 m/s was stated in Line 123. The wind speed has uncertainty of 32%, what do you quantitatively mean by "smaller contribution to the overall uncertainty"?

Line 284: Correct the spelling of "Sec.t"

Line 317f: Your argumentation about removal of NH3 seems invalid for two reasons: 1) Your downwind transect is 12 km downwind of the target CAFO, so with the reported wind speed the plume age is <24 minutes. NH3 has a lifetime on the order of hours and can be expected to still be present at that distance. 2) Your NH3:CH4 ratios span a wide range but are larger than reported in literature from NCFR studies (which are generally <50%), which would indicate that your NH3 background was set too low and/or CH4 background set too high.

Line 324f: I don't understand the conclusion here that it fails criteria 2. The SM ratio is the only one that compares to previous literature values that are based on observations for the NCFR reported in Table 2 in Kille et al. 2019.

Figure 5: I suggest shortening the caption. For panel a) descriptions are already included in the Figure and repetitive in the caption. For panel b), the two sentences in "Box plots of residuals for the four different methods. Residual boxplot of the different methods" are repetitive.

Lines 341-345: The wording is unclear, as line 344f states that Transect MVR is not the result with the lowest uncertainty. Then in the following sentence you state both MVR methods have lower uncertainty than SM. I suggest changing "This is, however, not the result with the lowest uncertainty" to "Transect MVR has slightly greater uncertainty than F2 MVR and both have lower uncertainty than SM because they do not require subtraction."

Line 351: "absolute errors and in quadrature" should be "absolute errors add in quadrature"

Line 370: Do you mean "of both CH4 and NH3" in this sentence?

Line 384: Missing grams in "2-12 g head-1 hr-1"

Line 386: Missing emissions in "The NH3 emissions are highly affected"

Data availability: Needs to be updated with link to dataset or explanation of how to access the data from the home page.

---

## Author Response (AR1)

We thank the referees for their meticulous responses which have enabled us to identify and correct inconsistencies in our results and provide a more thorough explanation of our methodology. We believe that the resulting manuscript is easier to understand and makes a more substantial contribution to the literature. We have addressed major comments by adding an additional case study from the full TRANS2Am campaign. In addition, we have more rigorously examined the potential sources in the region, provided new plots including a back trajectory plot in the SI and recalculated our results. We have also provided more support for our assumptions and evaluation of our attribution criteria. These major changes have required additional analysis that was completed by a graduate student who has been added to the author list for this work.

Reviewer comments are in bold with our responses below in plain text. Line numbers correspond to the final revised manuscript without tracked changes.

**Reviewer #1:**

**Specific Major Comments**:

**1) Concern about chosen data set: The publication by Pollack et al. (2022) utilized the exact same flight (F2) and performed similar analyses as reported in this manuscript such as the MVR and comparison of enhancement ratios. Pollack et al. (2022) also reported that the optimal flight track pattern would be racetrack or box patterns, whereas F2 used an approach that was determined to be less ideal (Lines 110-112, 113-115f of this manuscript). It is my understanding that TRANS2AM flights took place in 2021 and 2022 with such racetrack patterns flown (observed in UCAR field catalog). If this understanding is correct, I would suggest that flight data with a more optimized sampling strategy should be used as the basis for this technical note or an analysis of the errors resulting from non-ideal sampling should be included. My concern is that the results are not transferrable to analyses where the background of mixing ratios is more carefully determined from the racetrack patterns, which will cause unquantified biases, see comment 2) below.**

Thank you for bringing up this important point. The choice of flight track has implications for the type of analysis that can be done and what science objectives can be targeted (such as observing downsind NH3 depletion). However, we would not expect the choice of flight track and sampling strategy in this work vs the rest of the TRANS2AM campaign to have a significant impact on the accuracy of the results for emission quantification because ultimately the methodology is very similar. In response to your comment, we have decided to include data from the TRANS2AM flights with the same feedlot in the newly created Section 3.4, which utilized a more optimized racetrack pattern for sampling. This allows for a more comprehensive evaluation of the methods and results presented in this manuscript. However, we would like to clarify that a full analysis of the TRANS2AM data is beyond the scope of this technical note.

**2) Concern about secondary sources: You consider the background of the mixing ratios to be constant and use the transect that is 12 km downwind of the targeted CAFO to support that conclusion. This assumption, if incorrect, is substantially problematic. One way this assumption could be invalid is if you observe 2 plumes. In my understanding, the ideal flight pattern would be racetrack or boxes where you fully capture any upwind influence along a full transect that is transported towards the horizontal transect downwind, which would capture any secondary plume passing through your region of study. It appears to me looking at Figure 4 that you may have indeed measured two distinct sources contributing peaks in NH3 (and CH4), one near -10000m from center of**

**the F2 transect and one that is likely the targeted CAFO at +/- 5000m from the center of the transect. Additionally, C2H6 (and CH4) is enhanced from -10000 to the far end of the transect (on the right of the plot) but is not enhanced at the end of the transect towards the left (<- 10000m from center). Your conclusion in Line 384 is that the emissions per head from this study are higher than previous estimates. If there was another CAFO contributing to the second, smaller peak in Figure 4, then the per head emissions are higher than the actual per head emissions. Without this more ideal sampling strategy, how can you support your conclusion that you are observing an isolated source?**
Thank you for your comment. We have calculated NOAA HYSPLIT back trajectories for the two plumes mentioned in Figure 4, and have identified that the smaller plume is likely attributable to another nearby major CAFO. As a result, we have revised our cattle count to include the additional large CAFO near the target CAFO and the smaller CAFOs that may be added into the plume, which has lowered the estimated emissions to 13 ($\pm$3) g $CH_4$ head$^{-1}$ hr$^{-1}$ and 13 ($\pm$2) g $NH_3$ head$^{-1}$ hr$^{-1}$ and brought them closer to values reported in other literature.

Additionally, we acknowledge the importance of ideal sampling strategies in accurately isolating sources of emissions. As such, we have included data from the TRANS2AM flights in our analysis, which utilized more ideal flight paths close to the facilities and allowed for better isolation of feedlots.This flight also had multiple down wind transects. The final average estimates of 2022 flight emissions per head from this data set are consistent with those obtained from the 2019 flight. We believe that these results support the use of ideal sampling strategies in accurately identifying and quantifying emissions sources in this region with many potential sources.

**3) Concern about constraint for CH4 attribution: In Line 199 is stated that F2 is all data within the MBL during the flight – but your flight started and ended in Laramie, Wyoming (see Line 83), which is in a very different area than the observed CAFOs, separated from the NCFR by a mountain range. All flight data that is outside of a certain boundary of the NCFR should be excluded instead of only those outside a certain altitude range.**
Thank you for your comment. We agree that it is important to ensure that flight data from different areas is not combined in our analysis. To address this, we only used F2 data that was within the NCFR and excluded all flight data that was outside of a certain boundary of the NCFR. Specifically, we excluded any data from areas north and west of 41 degrees latitude, 105.25. This approach is similar to that used in the recent study by Pollack et al. (2022), which we have cited in our manuscript. This was done in our initial analysis and we added more detail to the manuscript on lines 90 -91 to clarify our methodology.

**Minor Comments**:

**Line 4: I suggest using present tense instead of past for "relied" unless the current method is something else. Then current method should be stated as well.**
Thank you for pointing this out. We have fixed it to the present tense on line 4 as this is the current method.

**Line 20: It is unclear whether the 8.4% growth is for the US or globe or a specific region within the US. Assuming it is the US based on second half of the sentence, I suggest "grew 8.4% in the US"**
Thank you for the suggestion, we made the suggested change on line 20 to be more clear.

**Line 22f: Remove second mention of EPA reference.**
We have made the revisions accordingly on line 23.

**Line 64f: This sentence is somewhat true but also misleading and weakening your conclusion in that MVR is the optimal method. I believe you are trying to say that using more variables in the MVR for additional sources can falsify the results and is not appropriate. On the other hand, using fewer variables in the MVR and only those you are certain that are sources will not falsify the results for those known sources – additional possible sources will be treated in the extra term or background. I would suggest adding to the sentence first a quick focus on how MVR is appropriate, then what situations make MVR inappropriate.**
We appreciate your suggestions for improving the clarity and accuracy of our discussion on the use of multivariate regression (MVR) for source attribution. We have revised the sentence on lines 66 - 69 in question to better reflect the appropriate use of MVR and the limitations of the method in certain situations.

**Line 66: Missing an "a" or "the" prior to "methodology"**
We have made the revisions accordingly on line 70.

**Line 78: Missing a space in "Greeley,CO"**
We have made the revisions accordingly line 82.

**Line 86: "COtextsubscript2" should be "CO2"**
We have made the revisions accordingly on line 91.

**Line 87: Suggest "This Picarro and other Picarro models" instead of "This model and other Picarro models" to avoid confusion with atmospheric model**
We have made the revisions accordingly on line 93.

**Figure 1: I suggest adding numbers for the middle two circles as well, as 1 and 125000 are the two extremes and it is nearly impossible to envision what capacity the intermediate circles represent.**
Thank you for your suggestion, we have made the revisions to the figure.

**Figure 1 caption: Are animal units determined by yourself or is there a reference? Are the equivalents with respect to CH4 or NH3 emissions or are the NH3:CH4 ratios between different animals identical? Please clarify.**
The animal units were taken from Colorado Department of Public Health and Environment (CDPHE,2017).

**Figure 1 caption: I suggest shortening the second sentence in the caption to "CAFOs are colored by animal type and sized by max animal units" as the color info is found in the legend.**
Thank you for the suggestion, we have made the revision.

**Line 125: Do you mean 6.5° variability in the wind direction? Which instrument was used for wind and what is its measurement uncertainty (0.65° as currently stated seems too low in my experience)?**
This was an error. However, not to confuse the reader, we have decided to omit it. We now state the wind speed and direction instrumental error on line (2.1 Data collection). The wind measurements were made using the aircraft 5-port gust probe, which is maintained by the UWKA team. A relevant publication has now been cited in the text on line 103 : https://doi.org/10.1002/qj.2604.

**Line 126: "at t 4 km- 14 km downwind" should be "at 4-14 km downwind"**
We have made the revisions accordingly on line 133.

**Line 191: Remove comma from "(2019),"**
We have made the revisions accordingly on line 198.

**Line 193, 196: Once you use ppb, once you use ppbv. Pick one and check throughout paper and figures for consistent use of units.**
Thank you for catching this, we are using ppbv as it is a volumetric unit. This was fixed on line 202.

**Line 201: "(ratio x 100)" should be "(ratio x 100%)"**
We have made the revisions accordingly on line 208.

**Line 228: "performed sensitivity analysis" should be "performed sensitivity analyses"**
We have made the revisions accordingly on line 241.

**Line 237: "regions for MVR analysis" should be "regions for MVR analyses"**
We have made the revisions accordingly on line 246.

**Line 238: I suggest moving Eq. 4 behind "as shown in Eq. 4." in Line 224 where it is described in the following sentence. Currently it seems to be floating without context prior to the next Section.**
Thank you for the suggestion. We have made the revisions accordingly; the equation now follows line 232.

**Line 246 and Figure 4: In Line 246 you state that you "screened out points that were <5 ppbv of NH3" but in Figure 4 I certainly seem to see cyan colors further to the left in the layer between 1500-2000 m AMSL as well as along the entire transect in the layer above 2500 m AMSL. The color bar indicates 5 ppbv is a darker shade than the cyan color.**
To clarify, we screened out points that were lower than 5 ppbv of NH3 based on the lowest altitude value in the transect. Once a value lower than 5 ppbv was determined in the lowest altitude, a vertical line was drawn to screen out points below this threshold. This approach was taken to ensure that only plumes associated with feedlots were included in the analysis, and not mixed with other unrelated plumes. Additionally, anything above 2600 msl was above the boundary and was removed. This explanation has been added to lines 255- 260.

**Figure 4: How were the vertical curtain bins determined? Why does the curtain not start at 1500 m AMSL, since you stated in Line 81 the lowest flight altitude was 100 m AGL and state in the caption that the surface is at 1400 MSL?**
We appreciate your question and have made some revisions to improve the clarity of the figure. The vertical curtain bins were determined based on the transects' altitude and are intended to represent the integrated area. We have updated the bins to more accurately reflect the full area of integration to the ground level. We would like to clarify that while the bins provide a visual representation of the data, they are not an exact image of the actual vertical distribution. Additionally, we have revised the caption to clarify that the surface is at 1400 MSL and the curtain bins start at that level.

**Table 1: Currently missing the CI superscript in the table header**
Thank you for catching this, we have added it back into the manuscript.

**Line 272: Wind speed of 8.4 +/- 2.7 m/s was stated in Line 123. The wind speed has uncertainty of 32%, what do you quantitatively mean by "smaller contribution to the overall uncertainty"?**
To clarify, we used the standard deviation of wind speed for each level of the transect, which is 1.3, 1.9 and 1.8 m/s for the lower, middle and highest altitudes, respectively. This was calculated into the perpendicular wind for the overall emission calculation. We found that the wind speed uncertainty had a relatively small impact on the overall uncertainty compared to other variables, such as background concentration. However, we understand that this may have been confusing, so we have removed the overall wind speed uncertainty from our discussion and instead included the instrumental uncertainty on lines 104 -105  to provide greater clarity for the reader.

**Line 284: Correct the spelling of "Sec.t"**
We have made the revisions accordingly on line 300.

**Line 317f: Your argumentation about removal of NH3 seems invalid for two reasons: 1) Your downwind transect is 12 km downwind of the target CAFO, so with the reported wind speed the plume age is <24 minutes. NH3 has a lifetime on the order of hours and can be expected to still be present at that distance. 2) Your NH3:CH4 ratios span a wide range but are larger than reported in literature from NCFR studies (which are generally <50%), which would indicate that your NH3 background was set too low and/or CH4 background set too high.**
There is considerable range in reported NH3 per head emissions in the literature and such ratios are also known to exhibit temperature dependence (See Table 4 Shonkwiler and Ham, 2018: https://doi.org/10.1016/j.agrformet.2017.10.031). Variability in NH3 per head emissions may translate to variability in NH3:CH4 ratios, which is part of the motivation of the TRANS2Am campaign. With regard to the possibility of NH3 deposition at close scales we note that this is also a motivator for TRANS2Am and an area being actively analyzed by our collaborators. Full exploration of these topics are outside the scope of this manuscript.
Regarding Point 1: There is support in the literature for deposition of NH3 even at close scales. For example, Staebler et al. 2009 (https://doi.org/10.1016/j.atmosenv.2009.08.045) observed some impact of deposition on the NH3 signal from aircraft flights within a few km downwind of a source. Miller et al. 2015 ( https://doi.org/10.1002/2015JD023241) showed significant impacts of deposition even within ~3 km of sources. We have added these references to the text. We have also added an additional case study from the TRANS2Am campaign collected in August '21 that shows clear evidence of decrease in NH3:CH4 ratio downwind of the site indicating NH3 is being depleted.
Regarding Point 2: Though our results for ratios are higher than some literature, they are in line with Pollack et al. 2022 which included observations from facilities other than the one reported in this study. Results from the same region in Golston et al. 2020 show a range of observations spanning from 1-200% across 213 facilities. Additionally, our methodology to calculate the NH3:CH4 ratios by MVR is independent of the CH4 or NH3 background as we showed in Section 2.5.2 and Tables S1-2.

**Line 324f: I don't understand the conclusion here that it fails criteria 2. The SM ratio is the only one that compares to previous literature values that are based on observations for the NCFR reported in Table 2 in Kille et al. 2019.**

As we have described in the response to the previous comment, there is a wide range of reported NH3:CH4 ratios in the literature. We have selected Pollack et al. 2022 as our primary reference to evaluate whether the results passed or failed criteria 2. Other work by Golston et al. 2020 (collected a few months before data in the Kille et al. 2019 study) in the same region showed a wide range of NH3:CH4 ratios by facility ranging from 1-200%. It is possible for individual facilities to have large ratios and still have a lower regional ratio. We have edited the text on lines 336-338 to include reference to the variable NH3:CH4 ratios in the literature.

**Figure 5: I suggest shortening the caption. For panel a) descriptions are already included in the Figure and repetitive in the caption. For panel b), the two sentences in "Box plots of residuals for the four different methods. Residual boxplot of the different methods" are repetitive.**
Thank you for the suggestion, we have made changes to the caption to shorten it.

**Lines 341-345: The wording is unclear, as line 344f states that Transect MVR is not the result with the lowest uncertainty. Then in the following sentence you state both MVR methods have lower uncertainty than SM. I suggest changing "This is, however, not the result with the lowest uncertainty" to "Transect MVR has slightly greater uncertainty than F2 MVR and both have lower uncertainty than SM because they do not require subtraction."**
Thank you for the suggestion. We have incorporated it into the paper for clarity on lines 362 -364.

**Line 351: "absolute errors and in quadrature" should be "absolute errors add in quadrature"**
We have made the revisions accordingly on line 370.

**Line 370: Do you mean "of both CH4 and NH3" in this sentence?**
Thank you for catching that. We have omitted that sentence from the paper as results have changed.

**Line 384: Missing grams in "2-12 g head-1 hr-1"**
We have made the revisions accordingly on line 441.

**Line 386: Missing emissions in "The NH3 emissions are highly affected"**
We have made the revisions accordingly on line 443.

**Data availability: Needs to be updated with link to dataset or explanation of how to access the data from the home page**
Thank you, we have added links for the publicly available data for both datasets on lines 475 - 479. The November '19 flight data has been submitted to WyoScholar, and is currently awaiting approval.

*Reviewer #2:*

*General Comments:* **This paper compares two methods to isolate CH4 emissions from feedlots from those of nearby oil and gas operations, using airborne measurements with NH3 as a tracer for the former and C2H6 for the latter. The multivariate method as proposed is shown to be superior to the subtraction method to which it is compared.**

**This comparison is a worthwhile exercise that will be of interest to the readership. Unfortunately, it is based on just one flight, which the authors admit had some design problems, primarily to do with potentially inadequate background concentration data. Reviewer 1 has elaborated extensively and with precision on these issues, so there is no point in belaboring them here. I second the idea of exploring other existing data sets, ideally from box flights rather than spirals, to further test the two methods. Formulating four criteria to evaluate the relative merits of the methods is also a laudable effort, but the relatively subjective manner in which this was executed was not very satisfying.**

Thank you for your valuable feedback. Based on your comments, we have included data from a second flight that took place during the TRANS2AM 2021 campaign and covered the same feedlot as the 2019 flight in the newly created Section 3.4. This flight had a more ideal flight track and provided us with additional data to compare the two methods. We found that the results for NH3 and CH4 were consistent with those from the 2019 flight. However, we acknowledge that a full analysis of the TRANS2AM data is beyond the scope of this manuscript. Regarding the four criteria we used to evaluate the methods, we appreciate your feedback and recognize that they may be subjective. We have revised the text to provide more clarity and transparency on how we applied the criteria and the limitations of this approach. We hope that these changes address your concerns and improve the quality of the manuscript.

**Specific Comments**

**Line 64/65: confusing sentence, please rephrase to clarify**

We have changed the end of the paragraph (lines 66-69) to better explain the appropriate situations when MVR should be used. We hope that this helps the flow  and will increase the reader's understanding.

**L86: CO2textsupscript2**

We have made the revisions accordingly on line 91.

**L124: that is a tiny variability on the wind direction, please check**

This was an error. To not confuse the reader, we have decided to report the instrumentation uncertainty of the wind instrument  on lines 102 -103  and explain the statistics used in the description of uncertainty analysis in Section 2.4.

**L138: define "limits of the plume width" quantitatively**

We have added clarification for how the plume width is defined using $NH_3$ on lines 151-152.

**L138: why project the plume height when you have data?**

We did use the available data to determine the plume height, but we wanted to clarify that we constrained (not projected)  the data to within the boundary layer to ensure accuracy. We have updated the wording in the manuscript to reflect this on line 153. Thank you for bringing this to our attention.

**L144: also worth checking the method described in Conley et al., https://doi.org/10.5194/amt-10-3345-2017**

Thank you for pointing out this reference. Conley et al., 2017 demonstrated spiral flight tracks can accurately estimate emissions, which is similar to our work. Their flight track was directly around the source, while we performed spirals around our target source and further downwind. We decided not to calculate emissions using the spirals due to the lack of information near the ground. While further downwind, one can assume the boundary layer is well mixed allowing for easier aircraft measurements. Secondly, the other main goal for the flight paths was to

characterize the NH3:CH4 ratios downwind of the source. This was not the focus for this paper, but rather, the focus for Pollack et al., 2022.

**L150: uncertainty section: Gordon et al. (https://doi.org/10.5194/amt-8-3745-2015) describe a comprehensive uncertainty analysis that may be worth investigating**
Thank you for pointing out Gordon et al. 2015. This paper demonstrates a box-like flight pattern near the source. Similarly to above, we chose to not use the near source spiral tracks because of the uncertainty near the ground. Instead our measurements are further downwind, where we can assume a well mixed boundary layer and therefore, reduce missed data near the ground. We have incorporated this reference in our mainscript on line 92.

**L168-183 can be significantly condensed**
We appreciate your suggestion to condense the text in L168-183, and we have made significant revisions to that section to improve its clarity and conciseness.

**L189-190 remove sentence**
We have made the revisions accordingly.

**L192: "instrument" precision**
We have made the revisions accordingly on line 201.

**L206: how do you justify this assumption? As far as I can tell, it is not strictly necessary anyway**
The literature (for example Kille et al. 2019) supports CAFOs and ONG activities being the dominant sources in this region accounting for ~90% of emissions. We have amended the text on lines 214-215 to include this rationale.

**L245: be more specific; perhaps "contiguous NH3 plume above 5 ppbv"**
Thank you for the suggestion, we made the correction on line 255.

**L273: 10% seems rather low.**
We would like to clarify that this value represents the overall error in the CH4 measurements for this particular experiment on this day, which had rather ideal conditions for sampling. We agree the uncertainty is rather low for this day, though not outside the realms of uncertainty reported in the literature (Golston et al., 2020; ). Emission uncertainty is a function of meteorological conditions and can change dramatically in non-ideal conditions such as highly variable backgrounds, shifting winds and quickly growing boundary layers.

**L284: Sec.t**
We have made the revisions accordingly on line 300.

**L276: extrapolation of concentrations to the ground level may introduce significant errors**
This is correct and a limitation of this methodology. While extrapolation to the ground may introduce errors, this is commonly used in other papers (E.A. Kort et al., 2018; ). Further, Cambaliza, et al. (2014) analyzed different techniques that extrapolate plumes to the ground and found that the comparison of different techniques did not produce a significant change in overall error. Our measurements were taken further downwind of the source

**L330-331: rather subjective approach to rejecting these results**

We agree that the rationale for rejecting the MVR results using all flight data is the weakest and have highlighted that this is the recommendation if transect specific MVR is not possible in lines 380-383. Our further criteria looking at goodness of fit statistics supports transect specific MVR.

**L369: if maximum cattle numbers are used, the already high emission factors found are likely underestimates.**
Thank you for your comment on the potential underestimation of emission factors when using maximum cattle numbers. We acknowledge that our approach may have limitations in accurately estimating emissions, particularly in cases where the maximum cattle head count is uncertain. While we recognize the limitations of this approach, it is a common practice in studies where the exact cattle count is not available, as noted in previous literature (Kille et al., 2017; Staebler et al., 2009; Laeytem et al., 2011).

**L378-383: a weak argument. Rather than speculating, it should be possible to find out on the ground whether significant feed modification is practiced in this location.-+++**
Thank you for your comment on our argument regarding feed modification practices in the study location. We acknowledge that obtaining ground-level data would be ideal, but given the practical constraints of our study, we were not able to directly work with the feedlots and therefore could not confirm or deny feeding operations. In general, we strive to minimize our direct contact with the farms to avoid any potential changes in their operations while the emissions are being measured.

**L389: Most airborne studies take place during the middle of the day, for logistical reasons**
While it is true that many airborne studies take place during the middle of the day for logistical reasons, an additional justification to conduct our measurements at midday is because this is when a fully mixed boundary layer occurs, which is optimal for our measuring technique.

**Fig. 3b: relationship is not clear from this figure. You could zoom in on the 1900-2000 ppbv range on the x-axis and show the > 2000 ppbv points separately & condensed**
We appreciate the feedback. Upon further consideration, we have decided to revise Figure 3b as suggested. We will zoom in on the 1900-2000 ppbv range on the x-axis and show the points with concentrations greater than 2000 ppbv separately and condensed for better clarity. Our intention with this revision is to better highlight the lack of a clear relationship between the variables.

**Fig 4: caption for d,e,f: Use wording similar to that suggested for L245, to be more precise, since there are also > 5ppbv NH3 at ~2400m around -13km, and near the center above 2500m. Panel b) does indeed suggest 2 sources, a smaller one near -10km, and a larger one in the center.**
Thanks for pointing attention to the confusion in the caption. We have updated the caption to include the explanation of points removed. Further, we performed NOAA HYSPLIT back trajectory calculations and identified two plumes. This has been updated in the manuscript.

**Fig. 5: please add units on the y-axes**
Thank you, this has been fixed on the manuscript.

**Fig. 5b: as it is, the only information clearly discernable are the ranges (error bars). You could dividing the range by 10**
We have decided to revise Figure 5b using a violin plot, as it allows for better visualization of the distribution and variability of the data. We have also zoomed into the zero line to highlight the

important variables. However, this approach may not display the ranges (error bars) as clearly. We added a note in the figure caption or in the results section explaining that the ranges are not displayed in the graph, but rather the outliers are out of frame. We believe that the use of a violin plot will provide a more informative and visually appealing representation of the data.

---

## Referee Report (RR1)

Second review of "Technical note: Isolating methane emissions from animal feeding operations in an interfering location" by McCabe et al.

May 9, 2023

Reviewer Recommendation:

Minor revisions before publication.

**Summary:**

The content of the submitted manuscript remains largely the same information as the first iteration of the article. Two methods to isolate CH4 emissions are described. The data is from airborne in-situ measurements. In addition to the original data from F2 now also data from RF13 is included to support the technical analysis. The changes made by the authors have clarified all my original concerns and focused the results adequately for publication.

I believe the revised manuscript is publishable, provided the authors address a few very minor comments that remain to be cleared up.

**Minor Comments:**

(Line numbers are with respect to the changes tracked document)

Figure 1 caption: missing space in "cattle= 2.5"

Figure 1 caption: suggest adding commas between the different animal head equivalents

Line 105: remove second space prior to "5 degrees"

Line 133: add m/s behind "windspeed of 8.4"

Line 134: the MBL figure is Fig. S2

Line 138: suggest using NH3 and CH4 for consistency

Line 156: bppv should be ppbv

Line 197: missing a minus in +/-

Line 282: remove "by" prior to or remove parentheses around "4%"

Figure 4 caption: the caption states "points on the left side of the curtain figure" – does this mean to the left of the vertically dashed line?

Figure 4 caption: Greely should be Greeley

Line 305: this sentence is incoherent, check grammar

Line 348: add parentheses around the two references

Line 350: add parentheses around the two references

Line 407: "of a ~1.5 hours" should be "of ~ 1.5 hours"

Table 2: Transect 5 distance should be 16.8 (as in Table S4)

Line 424: Monte Carlo should be capitalized

Line 457: "slightly than" should be "slightly higher than"